# Efficient Backdoor Mitigation in Federated Learning with Contrastive Loss

## Abstract

Due to the data-driven nature of deep neural networks and privacy concerns around user data, a backdoor could be easily injected into deep neural networks in federated learning without attracting the attention of users. An affected global model operates normally as a clean model in regular tasks and behaves differently when the trigger is presented. In this paper, we propose a novel reverse engineering approach to detect and mitigate the backdoor attack in federated learning by adopting a self-supervised Contrastive learning loss. In contrast to existing reverse engineering techniques, such as Neural Cleanse, which involve iterating through each class in the dataset, we employ the contrastive loss as a whole to identify triggers in the backdoored model. Our method compares the last-layer feature outputs of a potentially affected model with these from a clean one preserved beforehand to reconstruct the trigger under the guidance of the contrastive loss. The reverse-engineered trigger is then applied to patch the affected global model to remove the backdoor. If the global model is free from backdoors, the Contrastive loss will lead to either a blank trigger or one with random pattern. We evaluated the proposed method on three datasets under two backdoor attacks and compared it against three existing defense methods. Our results showed that while many popular reverse engineering algorithms were successful in centralized learning settings, they had difficulties detecting backdoors in federated learning, including Neural Cleanse, TABOR, and DeepInspect. Our method successfully detected backdoors in federated learning and was more time-efficient.

## 1 Introduction

In recent years, deep learning has achieved state-of-the-art results in various fields such as image classification (Karpathy et al., 2014), object detection (Karpathy & Fei-Fei, 2014), face recognition (Mehdipour-Ghazi & Ekenel, 2016), and self-driving cars (Bojarski et al., 2016; Grigorescu et al., 2019). Deep neural networks are driven by the vast amount of training data that may carry sensitive information about the users (Zhu et al., 2019). For privacy protection, the model owner often releases to the public and shared online only the trained model, but not the original training data. Recently, federated learning has been introduced to protect the user data privacy (McMahan et al., 2016). It allows the users to train a model locally and upload the model to the server, so that the server only needs to access and aggregate the uploaded models, without requiring access to the original local training data.

Despite its remarkable success, the federated learning scheme is becoming an increasingly attractive target for cyber criminals, due to the data sensitivity of deep neural networks, the transparency perspective of model training, and the publicly available nature of models that are shared online. These vulnerabilities have been unveiled and exploited by attackers to obtain representative information of user data (Zhu et al., 2019; Wang et al., 2018) or induce the model to produce improper results (Gu et al., 2017; Liao et al., 2018; Wang et al., 2020). Neural backdoor attack is a typical data poisoning attack that injects the backdoor into the model during training and it is only activated when a specific trigger is present. The poisoned model operates normally in regular tasks but differently when the trigger is present. The backdoor attack could potentially threaten the life of users in safety-critical applications such as autonomous driving, which has become an immense public concern.

To affirm the security and trustworthiness of deep neural networks, various state-of-the-art defense methods were proposed to identify and mitigate backdoor attack. One such method termed "Neural Cleanse" (Wang et al., 2019) was proposed to identify and mitigate the backdoor attack. It aims to reverse engineer the trigger by finding the input perturbation for each class that leads to the misclassification of the target class. DeepInspect (Chen et al., 2019) is the advanced version of Neural Cleanse that utilized a generator to find the backdoor. On the other hand, the Artificial Brain Stimulation (ABS) method (Liu et al., 2019) analyzes internal neurons in a suspicious model by stimulating different inputs into the model and testing the activation outputs of the neurons to identify the compromised ones. An optimization method is then applied to these compromised neurons to reverse engineer the trigger. TABOR (Guo et al., 2019) is another method that was designed to solve a complex objective function to detect the backdoor and reverse engineer the trigger. DeepSweep (Zeng et al., 2020) cleanses input data for the model during training, Activation-clustering (Chen et al., 2018) utilizes the activation outputs of the model to identify and remove backdoor, and NeuronInspect (Huang et al., 2019) generates heat maps from the output layer of the model to detect backdoor. However, these defense mechanisms have various drawbacks in terms of effectiveness, time efficiency, and computational costs, which limit their application in federated learning.

To tackle those challenges, we propose a novel approach capable of detecting and unlearning the backdoor in large-scale federated learning systems. Our method utilizes a clean, pretrained model and adopts a self-supervised learning contrastive loss to efficiently reverse engineer the backdoor trigger. While existing defense mechanisms in federated learning (Pillutla et al., 2019; Fung et al., 2018) involve examining updates from local models or modifying the aggregation algorithm to ensure the safety of the global model, our method directly targets the global model without the need to inspect local updates or change the aggregation scheme.

The contrastive loss (Dey et al., 2017; Chopra et al.; Koch, 2015) is typically used to map similar/dissimilar patterns in input into the same/different locations in the feature embedding space. In our self-supervised contrastive learning setting, we first inject some perturbations into an image to mimic the backdoor trigger, and let the poised image go through the clean and suspicious models. We then utilize the contrastive loss (Chopra et al.; Neculoiu et al., 2016; Chen et al., 2020) to train the perturbations so that the difference between the clean and suspicious models is maximized. Our intuition is that a backdoor trigger usually has a fixed pattern and it can generate strong responses at certain neurons in the model, leading to misclassifications of images of whatever classes to the target class once the trigger is presented. The Contrastive loss will guide the optimization to adapt the perturbations to match the trigger, if the model is backdoored, because only the trigger can produce such strong responses. If the suspicious model is clean, only random noise or a blank image trigger can be reconstructed, and the difference between the outputs of the two models is small.

To evaluate our method, we tested it on three datasets, with three models and two types of backdoor attacks. We compared the results with three well-known reverse engineering trigger algorithms. The experimental results showed that our method efficiently mitigated backdoors in federated learning where popular reverse engineering methods failed, achieving fast and effective protection against backdoor attacks. This is particularly important for federated learning, since the mitigation scheme must be applied in every round of model aggregation. As the number of classes and model complexity increase, the advantages of our method over other competing algorithms grow accordingly, because we do not iterate each class. Our main contributions are:

- We propose a novel method adopting contrastive loss to efficiently detect and mitigate backdoor in federated learning without inspecting local model updates or accessing local training data.

- Our method does not iterate through each class in the dataset to generate the trigger, nor does it require complex internal neuron analysis, heavy computational resources, or a huge amount of data resources.

- Our method can be directly applied to remove backdoors without affecting the aggregation schemes used in federated learning. After the backdoor is removed, the resulting models maintain high accuracies.

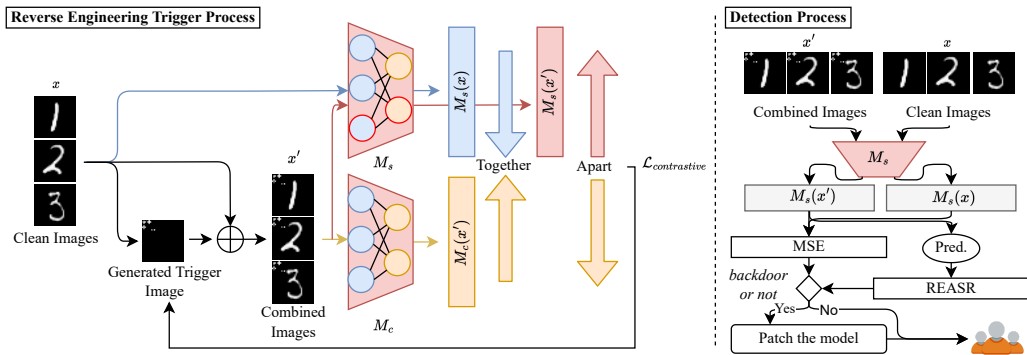

Figure 1: The overview of the proposed method (left) and the backdoor detection process (right). Left: $M_c$ and $M_s$ represent the clean and suspicious models, respectively, while $x$ and $x'$ represent the clean and combined images. The combined image is obtained by merging the generated trigger image with the clean image. We create positive feature representation pairs $M_c(x')$ and $M_s(x)$, as well as negative pairs $M_c(x')$ and $M_s(x')$. During training, the contrastive loss is used to guide the trigger image generation process, so that the positive pairs are brought closer together in the feature space while the negative pairs are pushed further apart. Right: we compute two metrics: $MSE = E_{x,x'}[||M_s(x') - M_s(x)||_2^2]$ and $REASR$ (reverse engineered attack success rate), which is calculated by using $M_s$ to classify a batch of combined images. If both metrics are above pre-defined thresholds, $M_s$ is determined as a backdoor model (See Section. 2.5).

## 2 METHODOLOGY

### 2.1 THREAT MODEL

We assume that there exist one or multiple attackers in a federated learning network who inject a backdoor during local training and upload its weight updates to poison the global model through FedAVG aggregation. Backdoor detection can be carried out either by the server or by local users. In either case, the server or the local users have a clean historical model with reasonable performance. This clean model could be a previously verified old version of the global model, or alternatively, a pre-trained model from a trustworthy party such as the Pytorch library. This clean model is utilized to establish the contrastive loss. The server or the local users have a very small set of clean data to sanity-check the model in hand. The sanity-check dataset is used for basic tests to quickly validate that the performance of the model is acceptable. These assumptions are appropriate and reasonable due to the fact that organizations in Federated Learning are typically cooperative and often possess a pre-trained or historical global model with satisfactory performance for specific tasks.

### 2.2 OVERALL FRAMEWORK

We introduce a novel method to reverse engineer the trigger and utilize it to remove the backdoor knowledge from the global model by patching the model with the generated trigger. We treat the trigger construction process as an optimization problem, where the trigger image is updated by comparing the feature outputs of the clean model and the suspicious model through THE contrastive loss. Fig. 1 shows the overview of our method, where the clean model ($M_c$) and suspicious model ($M_s$) have the same architecture.

### 2.3 LOSS FUNCTIONS

Backdoor knowledge is the main difference between the two models. To capture it, we utilize cosine similarity as our proxy to discover the difference between the two models. The difference is represented in the image space as trigger or perturbation, which causes the two models to behave differently. The main loss function comprises three terms. The first term is the adaptive contrastive

loss,

$$\mathcal{L}_{contrastive} = -\log \frac{\exp\left(sim\left(M_c(x'), M_s(x)\right)\right)}{\exp\left(sim\left(M_s(x'), M_c(x')\right)\right)}, \tag{1}$$

where $x$ denotes a clean image, $x'$ represents the combination of the clean image with the generated trigger image. $M_c$ is the clean model and $M_s$ is the suspicious model. We create positive feature representation pairs $M_c(x')$ and $M_s(x)$, as well as negative pairs $M_c(x')$ and $M_s(x')$. During training, the contrastive loss is used to guide the trigger image generation process, so that the positive pairs are brought closer together in the feature space while the negative pairs are pushed further apart.

To ensure that the generated trigger image does not significantly alter the structure of the original image, we include the Structural Similarity Index Measure (SSIM) (Wang et al., 2004) as the second term of our loss function,

$$\mathcal{L}_{SSIM} = -SSIM(x, x'), \tag{2}$$

SSIM measures the similarity in luminance, contrast, and structure of two images. By incorporating SSIM, we can ensure that the combined image maintains the structure of the clean image while still incorporating the backdoor trigger. In order to evade detection and maintain the stealthiness of the backdoor attack, the attacker typically injects a backdoor trigger into clean images while not significantly altering the structure of the clean images. Therefore, poisoned images are similar to clean images in general.

To ensure that the generated trigger image is not too large to significantly alter the structure of the original image, we include an $L_1$ norm as the third term of our loss function,

$$\mathcal{L}_1 = \|m\|_1, \tag{3}$$

where $m$ is the mask that is used to combine the trigger and clean images. The $L_1$ norm measures the absolute summation of each element in a vector. By including this regularization term in our loss function, we can limit the size of the generated trigger and prevent it from dominating the image.

The final loss function is formulated as a combination of the three components above, each weighted by a hyperparameter that controls the degree of regularization. Specifically, the final loss function is given by,

$$\mathcal{L}_{final} = \mathcal{L}_{contrastive} + \alpha\mathcal{L}_{SSIM} + \beta\mathcal{L}_1, \tag{4}$$

where $\alpha$ and $\beta$ are hyperparameters that control the trade-off between the different terms of the loss function. By adjusting these hyperparameters, we can fine-tune our method and achieve the best performance for a given task.

## 2.4 MOTIVATION OF THE LOSS FUNCTIONS

The overall loss function consists of three components. The contrastive loss, $L_{contrastive}$, captures the difference in feature space generated by the trigger between the clean model and suspicious models. The $L_{SSIM}$ loss ensures the trigger image does not significantly change the structure of the clean image, and the $L_1$ loss restricts the size of the generated trigger. During the reverse engineering process, the algorithm attempts to search for a unique pattern (trigger) that maximizes the difference between the negative feature representation pairs and minimizes that between the positive pairs. If the suspicious model contains a backdoor, the search process can find the trigger, as the true trigger is static with a fixed pattern and the backdoor planting process results in convolution kernels that are strong enough to cause misclassification for any images embedded with the trigger. In contrast, if the suspicious model is backdoor-free, the contrastive loss for any trigger should be small since the search process won't be able to find a match among the convolution kernels in the suspicious model.

In this study, we assume that the attacker aims to create a backdoor that is effective and stealthy, while minimizing the impact of the trigger on the clean images and keeping the size of the trigger small. The $L_{SSIM}$ loss ensures that the trigger does not significantly alter the structure of the clean images, while the $L_1$ loss constraint the size of the trigger. In summary, the loss function is designed to precisely recover a small and stealthy trigger that can be used to remove the backdoor knowledge from the backdoor model without significant effects on regular tasks, i.e., maintaining high classification accuracies on clean images.

## 2.5 BACKDOOR DETECTION

After generating the trigger, we combine it with a batch of clean images and feed both the clean and combined images to the suspicious model, as shown in the right part of Fig. 1. We then use two metrics to detect if the model has been backdoored. Firstly, we calculate the mean squared error ($MSE$) between the features outputted by the suspicious model for the clean and combined images. If the $MSE$ exceeds a certain threshold, it suggests the model has a backdoor, since a backdoored model will strongly respond to the trigger image. Secondly, we calculate the reversed engineering attack success rate ($REASR$) of the model on the combined images. This is achieved by computing the ratio of the model's predictions on the combined images to the mode of the predictions. The intuition is that if a model is backdoored and has a high attack successful rate during training by the attackers, it will misclassify most of the combined images as the target class, which is the "Mode" of the predictions. If both $MSE$ and $REASR$ are above the predefined thresholds, we determine the model as backdoored, and otherwise, we distribute it directly back to the clients. We do not perform any complex statistical analyses to determine the thresholds, and further details are available in the implementation section.

## 2.6 BACKDOOR UNLEARNING

If the suspicious model is identified as a backdoor model, we employ the machine unlearning technique similar to Neural Cleanse (Wang et al., 2019) to remove the backdoor knowledge through fine-tuning. In this process, we combine a portion of the clean images from batches of the sanity dataset with the generated trigger to form a set of poisoned samples. These combined images are assigned with the correct labels of the original sanity dataset and mixed with clean images. Then it would be utilized to fine-tune the backdoor model. After fine-tuning, the backdoor model is expected to maintain its performance on clean data while correctly classifying the poisoned images. In our experiments, we fine-tune the backdoor model for only one epoch.

## 2.7 PERFORMANCE METRICS

We adopt five metrics in our experiments to evaluate our method. Accuracy ($Acc.$), which measures the model accuracy on clean data. Attack Success Rate ($ASR$), which measures the model accuracy of the combined images successfully misclassified to the target label. The combined images refer to clean images stamped with the constructed trigger. $Speedup$, which measures the relative performance of two methods on reverse engineering the trigger. It is calculated as the ratio of the performance of the Neural Cleanse over the performance of the utilized method. The higher the $Speedup$ value, the faster it is in comparison to the baseline method Neural Cleanse. Mean Squared Error ($MSE$), which measures the average squared difference/distance between the latent outputs of combined images and clean images in the model. $REASR$, which measures the model accuracy of the combined images misclassified to the target class. The latter two metrics are for the backdoor detection purpose in our study.

## 3 EXPERIMENT SETUP

### 3.1 DATASETS AND MODEL ARCHITECTURES

To evaluate our method, we utilized three image classification datasets: MNIST (Deng, 2012), CIFAR10 (Krizhevsky & Hinton, 2009), and GTSRB (Houben et al., 2013). For these datasets, we utilized a deep convolutional neural network (ConvNet), ResNet18 and ResNet32 (He et al., 2015), the tasks of each model were for handwritten digits classification, object recognition, and traffic sign recognition, respectively. The implementation was conducted with NVIDIA GeForce RTX3080. The dataset and model architecture information are listed in Table 1. The detail is in Section A.

### 3.2 FEDERATED LEARNING PARAMETERS

Federated learning enables multiple parties to collaboratively train a model without sharing their private data, thereby preserving data privacy and reducing communication costs (McMahan et al., 2016). It is also more difficult for attackers to perform a backdoor attack in a federated learning

Table 1: Specifications of datasets and model structures

| Dataset | # Training Samples | Input Size | # Classes | Model Name | Model Parameters | # Local Epochs / # Rounds |
|---------|-------------------|-----------|-----------|------------|-----------------|--------------------------|
| MNIST | 60,000 | 1x28x28 | 10 | ConvNet | 102,730 | 3 / 60 |
| CIFAR10 | 50,000 | 3x32x32 | 10 | ResNet18 | 11,173,962 | 12 / 140 (SBA), 80 (dba) |
| GTSRB | 39,209 | 3x32x32 | 43 | ResNet32 | 21,306,731 | 20 / 60 |

scenario, as planting or detecting a backdoor is typically more challenging. To speed up the model training process and ensure high attack success rates, we set the number of total clients to 20 and randomly selected six clients at each round to send their local gradients to the server for model updating by FedAvg. The server then distributed the updated model back to all clients. The training dataset was randomly split among all clients to ensure it was independently and identically distributed. The global model was trained until it reached the desired performance level for a given task. The Adam optimizer was used for training the model.

## 3.3   TRIGGER PATTERN DESIGN

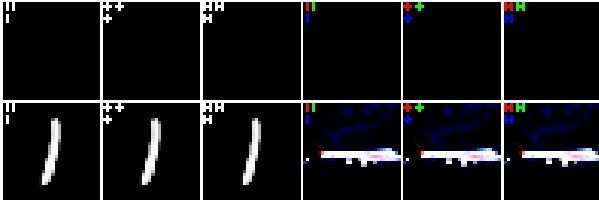

Figure 2: Designed triggers (first row) and combined images (second row). The gray scale triggers in the first three columns are used in the single backdoor attack (SBA), the color triggers in the last three columns are used in the distributed backdoor attack (DBA), except for MNIST dataset.

We designed three triggers consisting of three different atoms, as shown in Fig. 2. These atoms include the digit one ("1") composed of 3 pixels, the plus sign ("+") composed of 5 pixels, and the letter ("H") composed of 7 pixels. A complete trigger contains three atoms and is either a grayscale or a color image with a possible size of 9, 15, or 21 pixels, respectively. Fig. 2 shows all the triggers.

## 3.4   ATTACKING SCENARIOS

We implemented two attacking scenarios, including the Single Backdoor Attack (SBA) and the Distributed Backdoor Attack (DBA). In SBA, one selected participant used a complete grayscale trigger to plant a backdoor (Fig. 2, left). For example, the attacker combined the trigger with a batch of clean images and labelled the combined images as a chosen target class. After training, the attacker uploaded the gradients to the server for model update. The attacker later used the same complete grayscale trigger to activate the backdoor. In DBA, three selected attackers used an atom of the complete color trigger (Fig. 2, right) to plant a backdoor. The attackers combined an atom of the trigger with a batch of clean images and labelled them as a chosen target class. To activate the backdoor, the attackers used the complete color trigger. In our experiment, we simulated our backdoor attack based on BadNet (Gu et al., 2017) and the DBA (Xie et al., 2020).

## 3.5   DETAILED BACKDOOR DETECTION PROCEDURE

We assume that a clean model $M_c$ is available to us, either from a previously verified model or validated by a third party. To detect if a suspicious model $M_s$ is backdoored, we first randomly initiated two variables, the mask ($m$) and the trigger image ($T$), multiplied them, and combined the result with a batch of clean images to form the combined images ($x'$). The clean images $x$, together with $x'$, were then fed into the model (Fig. 1, left) to generate the trigger image through the proposed reverse engineering process. Once the trigger image was generated, we used the proposed detection process (Fig. 1, right) to determine if $M_s$ was backdoored. If it was, we utilized the combined

images $x'$ with the correct labels to fine-tune $M_s$. In our experiment, the number of clean samples chosen was three times the number of classes in the dataset. See Algorithm at section A.

## 4 RESULTS AND ANALYSIS

### 4.1 RESULTS OF BACKDOOR PLANTING

We first planted backdoors in the models we selected for this study. We have three different model structures and three datasets listed in Table 1, and three trigger image patterns shown in Fig. 2. In SBA, we used the triggers in the first three columns, while in DBA, we used the triggers in the last three columns of the figure. We conducted a backdoor planting experiment per dataset, per trigger, and per attacking scenario, and the average performances are listed in Table 2, where "Acc." denotes the accuracy of a model on clean data, and "ASR" represents the attack success rate of a backdoored model, measuring the ratio of the combined images being classified as the target class. It is observed that all backdoored models performed well on clean images but also can be activated at very high ratios.

Table 2: Backdoor planting results

| Dataset | Base Model | | Backdoor Model | | | |
| --- | --- | --- | --- | --- | --- | --- |
| | | | SBA | | DBA | |
| | Acc. (%) | ASR (%) | Acc. (%) | ASR (%) | Acc. (%) | ASR (%) |
| MNIST | 96.51 | 1.15 | 98.72 | 99.66 | 98.60 | 100.0 |
| CIFAR10 | 88.61 | 2.34 | 92.46 | 98.27 | 91.14 | 99.98 |
| GTSRB | 98.97 | 0.03 | 99.85 | 88.08 | 99.36 | 100.0 |

### 4.2 REVERSE ENGINEERED TRIGGER IMAGES

Fig. 3 displays samples of the generated trigger images by each method in the experiment. The intention was to imprint these images onto the clean images, creating combined images that would effectively trigger the backdoor model with a high attack success rate. The quality of the generated trigger images directly impacts the effectiveness of the fine-tuning process and ultimately improves the precision of the metrics used to detect the backdoor in the model. We observe that the generated trigger images by Neural Cleanse and Tabor have a lot of noise. DeepInspect generated a clear image showing the pattern in the proper location, whereas our method clearly reconstructed the atom in SBA. In row 2, which is the DBA setting, the attack is more robust, and all methods started to have difficulty reconstructing the trigger image. However, our method was still able to use the generated images to effectively reduce the backdoor knowledge of the backdoor. Table 3 shows the results after we applied these generated images to fine-tune the backdoored models.

### 4.3 RESULTS OF BACKDOOR DETECTION

We conducted backdoor detection using the proposed detection procedure as shown in Fig. 1 and results are shown in Fig. 4, where blue bars represent $MSE$ results and red bars denote $REASR$ results. For each dataset, both metrics were computed for clean, SBA attacked, and DBA attacked models, respectively. Each model was attacked three times with the three trigger atoms and averaged metrics values were computed. It is observed that there are significant margins in both metrics between clean and backdoored models, regardless the types of the backdoor attack. To further demonstrate the effectiveness of our method, we applied it to 180 models, half of which were clean, and half were backdoored, for backdoor detection. We used the same settings as those in Section 2.5 and A.2, and we set the thresholds to 10 and 80 for $MSE$ and $REASR$, respectively, to identify backdoored models. Table 4 presents the detection results, and our method only lost one out of the six cases.

Table 3: Results of patched models and computational efficiency of different methods

| Dataset | Method | Patched Model | | | | Trigger Reverse Engineering | | | |
|---|---|---|---|---|---|---|---|---|---|
| | | SBA | | DBA | | SBA | | DBA | |
| | | Acc.(%) | ASR(%) | Acc.(%) | ASR(%) | Time(s) | Speedup | Time(s) | Speedup |
| MNIST | NC | 98.49 | 5.90 | 98.33 | 88.22 | 35.96 | 1.00 | 35.38 | 1.00 |
| | TABOR | 98.48 | 23.45 | **98.44** | 99.99 | 56.28 | 0.64 | 55.47 | 0.64 |
| | DeepInspect | **98.54** | 5.35 | 98.36 | 99.15 | 78.98 | 0.46 | 78.56 | 0.45 |
| | Ours | 98.40 | **1.73** | 97.95 | **7.41** | **7.38** | **4.88** | **7.63** | **4.64** |
| CIFAR10 | NC | 90.18 | 57.77 | 87.83 | 5.32 | 77.25 | 1.00 | 76.62 | 1.00 |
| | TABOR | 90.28 | 21.33 | 87.36 | 16.56 | 213.59 | 0.36 | 215.36 | 0.36 |
| | DeepInspect | 90.48 | **0.89** | **88.48** | 3.25 | 141.15 | 0.55 | 140.18 | 0.55 |
| | Ours | **90.83** | 0.94 | 88.18 | **1.25** | **27.88** | **2.77** | **31.34** | **2.44** |
| GTSRB | NC | 99.42 | 23.50 | 99.29 | 42.16 | 872.22 | 1.00 | 880.40 | 1.00 |
| | TABOR | 99.30 | 60.57 | **99.34** | 31.86 | 1501.72 | 0.58 | 1522 | 0.58 |
| | DeepInspect | 99.42 | 41.08 | 99.21 | 24.63 | 922.90 | 0.95 | 932.29 | 0.94 |
| | Ours | **99.45** | **0.003** | 99.16 | **0.31** | **32.08** | **27.19** | **33.08** | **26.62** |

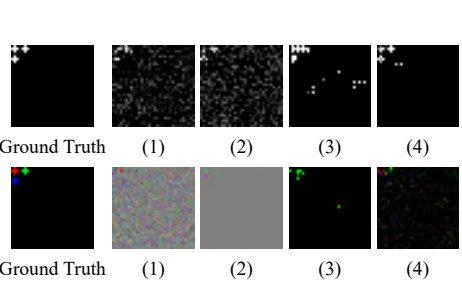

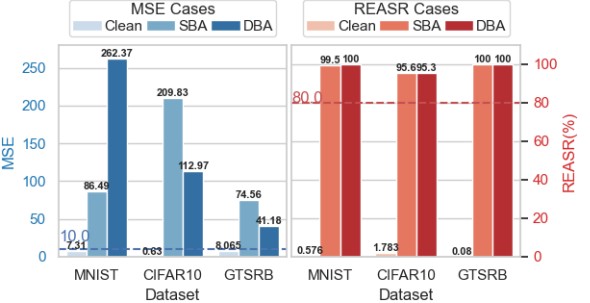

Figure 3: Reverse engineered trigger images by (1) Neural Cleanse, (2) Tabor, (3) DeepInspect, and (4) Ours in SBA setting (first row) and in DBA setting (second row), respectively. The atom of the ground truth trigger image is the plus sign $(+)$.

Figure 4: MSEs and REASRs of clean and backdoored models.

## 4.4 RESULTS OF BACKDOOR PATCHING

In our experiment, we patched the backdoored models by fine-tuning them using the generated trigger image. We compared our method with three reverse engineering trigger algorithms, including NC (Wang et al., 2019), Tabor (Guo et al., 2019), and DeepInspect (Chen et al., 2019). These existing methods evaluate each of the classes one at a time, use optimization methods to generate potential triggers, and utilize the Median Absolute Deviation (MAD) metric to identify the trigger. We adopted the implementations from the Trojanzoo backdoor library (Pang et al., 2022; 2020a;b) and used their default settings, except that the number of epochs was set to 500, and the number of samples was set the same as that for our method for fair comparison. Table 3 shows the experimental results of the proposed method and three competing algorithms, where all results are the averaged performances for the three triggers. If we set ASR below 20% after the patching procedure as a failure case, the three competing methods failed 5 out of 9 cases combined, while our method succeeded in all three attempts in the SBA attacking scenario. In the DBA attacking scenario, the three competing methods failed 6 out of 9 cases, while our method succeeded in all three cases.

## 4.5 COMPUTATIONAL EFFICIENCY

To compare the computational efficiencies of the competing methods, we used the computation time needed by Neural Cleanse for trigger reverse engineering as the baseline and computed the "Speedup" of the time by other methods as compared to the baseline. Therefore, Neural Cleanse has a "Speedup" of 1, and the larger the "Speedup" is, the more efficient the method is. Table 3 lists

the "Speedup" for each of the competing methods, clearly indicating the efficiency of the proposed method. The more classes the dataset contains, the more efficient our method is.

## 5 DISCUSSIONS AND LIMITATIONS

We investigated the impact of trigger size and the number of clients on the performance of our method and found that, while the proposed method can effectively detect the backdoor if the trigger size is smaller than $8 \times 8$ pixels, the impact of the number of clients in the federated learning setting is negligible. In addition, we found that the accuracy of the clean global model could be as low as 71% to achieve good backdoor detection performance. More details can be found in Section A. The key innovation of our method is that it does not iterate through each class in the dataset, as in the case of Neural Cleanse. Instead, it looks at the contrastive loss as a whole to search for triggers in the backdoored model. This novel method potentially provides two advantages: 1) it is super-sensitive in the federated learning setting, where traditional methods have failed (Table 3), and 2) it has the potential to speed up the search process, especially for datasets with a large number of classes. We have successfully demonstrated the proposed algorithm on several image datasets with 10-43 classes. We believe it can be applied to many other applications with a wide range of classes and we will evaluate their efficiency in our immediate future work.

## 6 RELATED WORK

**Federated learning.** It enables multiple parties to collaboratively train a machine learning model without sharing their private data with each other or with a central server, thereby preserving data privacy and reducing communication costs. FedAvg proposed by McMahan et al. (2016), is the fundamental aggregation algorithm in Federated learning, it reduces the number of communication rounds required for training, and allows users to train the model locally over multiple iterations. The existing defensive mechanisms (Pillutla et al., 2019; Fung et al., 2018; Yin et al., 2018; Sun et al., 2019) in federated learning were developed using the FedAvg scheme as a baseline.

**Backdoor attack.** It was first introduced by Gu et al. (2017) in centralized learning. An attacker first poisons a portion of training data by altering one or a group of pixels ("trigger") in a set of training images and set their ground truth labels as a chosen target class. They then train the model with the altered images with clean data to ensure that the backdoored model can maintain its accuracy on clean images and misclassify any images to the target class if the trigger is presented. Later on, the backdoor attack (Bagdasaryan et al., 2018; Wang et al., 2020) was extended to federated learning.

**Backdoor defense.** To defend against the backdoor attack, Wang et al. (2019) proposed the Neural Cleanse method. It examines each class separately and uses optimization methods to search for input perturbations that cause misclassifications. The Median Absolute Deviation (MAD) metric is then employed to identify triggers from the resulted perturbations. Another technique, DeepInspect (Chen et al., 2019), utilizes a conditional generator to reconstruct triggers for each class and performs statistical analysis to identify the backdoor. Additionally, TABOR (Guo et al., 2019) was proposed for trigger reconstruction.

**Contrastive learning.** It is a self-supervised method that learns the feature representation of the input data without labeling the dataset. Our work is inspired by the self-supervised contrastive learning method SimCLR (Chen et al., 2020). SimCLR utilizes positive and negative pairs to advise the model to learn the feature representation of the dataset. It maximizes the similarity between the positive pairs and minimizes the similarity between negative pairs.

## 7 CONCLUSION

This paper proposed an innovative approach to mitigate backdoor attacks in federated learning. Our method leverages the contrastive loss with a trustworthy historical model to reverse engineer the trigger. Experimental results showed that the proposed method can effectively mitigate backdoors as compared to Neural Cleanse, TABOR, and DeepInspect, which were very successful in centralized learning settings but faced tremendous challenges in federated learning. Our results indicate that maintaining a clean global model at hand could be a valuable practice.

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

## A  APPENDIX

### A.1  DATASET, MODEL ARCHITECTURE AND EXPERIMENT SETUP

The MNIST dataset contains 60,000 black and white handwritten digit images for training and 10,000 images for testing, each image has a size of 28x28x1 belong to one of the 10 classes. The CIFAR10 dataset contains 50,000 color training images and 10,000 testing images, each with a dimension of 32x32x3. There are 10 classes in this dataset. The German Traffic Sign Recognition Benchmark (GTSRB) dataset consists of 39,209 training images and 12,569 test images, with each image having a size of 32x32x3. The total number of classes for this dataset is 43. The deep convolutional neural network (ConvNet), ResNet18 and ResNet32 (He et al., 2015) models were utilized. The ConvNet network is comprised of four convolutional layers and a fully connected layer. Each convolution layer follows a LeakyReLu activation function and a dropout layer. For the ResNet models, we modified the last output layer to match the number of classes in the datasets.

### A.2  HYPERPARAMETER OPTIMIZATION

Based on trial and error, we used different values for the hyperparameters $\alpha$ and $\beta$ in our experiments. Specifically, for MNIST with SBA and DBA setting, we set $\alpha$ to 0.5 and $\beta$ to 0.01 for the loss function. For CIFAR10, we used 0.001 and 0.0005 for $\alpha$ and $\beta$, respectively, and for GTSRB, we set them to 0.001 and 0.005. During the reverse engineering trigger training, we fixed the number of epochs to 500 and used 30, 30, and 129 clean images for MNIST, CIFAR10, and GTSRB, respectively. For the unlearning or fine-tuning process, we set the number of iterations to 1. In our experiments, we used the default hyperparameter settings for the competing methods as specified in the Trojanzoo library (Pang et al., 2022; 2020a;b), but the number of epochs and the number of images followed our settings.

### A.3  BACKDOOR TRIGGER REVERSE ENGINEERING ALGORITHM

---
**Algorithm 1** Backdoor Trigger Reverse Engineering Algorithm

---
Input: clean images $x$, combined images $x'$, pre-trained clean model $M_c$, suspicious model $M_s$, a batch of clean data $B$ from sanity dataset $D$
Output: Mask $m$, Trigger $T$

1: Initialize $m$ and $T$ with random noise.
2: Sample a subset of clean images $x \subseteq B$.
3: **while** $epoch \leq num\_epochs$ **do**
4:     $x' \leftarrow m \oplus T + (1 - m) \oplus x$ .
5:     $L_{contrastive} \leftarrow -log \frac{exp(sim(M_c(x'), M_s(x)))}{exp(sim(M_c(x'), M_s(x')))}$.
6:     $L_{ssim} \leftarrow -SSIM(x, x')$.
7:     $L_1 \leftarrow \|m\|_1$.
8:     $L_{final} \leftarrow L_{contrastive} + L_{ssim} + L_1$.
9:     Update $m$, $T$ to minimize $L_{final}$.
10: **end while**
11: return $m$, $T$

---

### A.4  IMPACT OF TRIGGER SIZE, NUMBER OF CLIENTS AND ACCURACY OF BASE MODEL

We utilized the Single Backdoor Attack (SBA) with the CIFAR-10 dataset in the context of federated learning to investigate the impacts of trigger size, the number of clients, and the selection of historical clean model on the performance of the proposed method.

**Impact of trigger size.** To investigate the impact of trigger size, we designed a white square trigger located at the bottom right of the image. The trigger sizes are 2x2 (0.3% of the image area), 4x4 (1.6%), 8x8 (6%), and 12x12 (14%), as shown in Fig.5. Figs.6 a) and b) showcase that our method worked well for small trigger sizes (2x2 and 4x4), where the backdoors were successfully mitigated (ASRs close to 0 after fine-tuning), while accuracies for clean images remained high. The reason is that there is one $\mathcal{L}_1$ norm component in the loss function that restricts the trigger size to be small.

We assume the attacker uses small triggers to make the attack stealthy, which is consistent with these assumptions by defense mechanisms like Neural Cleanse. The proposed method does not target stealthy attacks with large triggers, such as invisible attack (Li et al., 2021). Note that in this experiment, we used squared triggers. The proposed method also worked well for slightly larger triggers (21 pixels) of different shapes, as shown in Fig. 2.



| 2x2 | 4x4 | 8x8 | 12x12 |

Figure 5: Designed white square triggers in various sizes

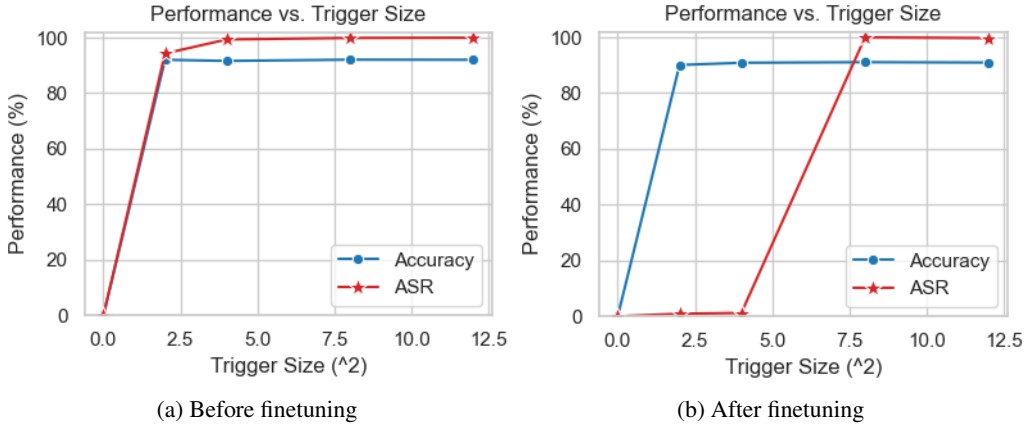

(a) Before finetuning

(b) After finetuning

Figure 6: Impacts (ASR and Accuracy) of trigger size on the performance of the proposed method.

**Impact of the number of clients in federated learning.** To evaluate the impact of the number of clients on the performance of our method, we used a trigger set of '1'. In the scenario with 50 clients in federated learning, the attack started at round 56 of weight updating, with a total of 150 rounds and an active client count of 15 per round. In the case of 100 clients, the attack started at round 93 out of a total of 250 rounds, with 30 active clients in each round. Fig. 7 illustrates that our method is robust to the number of clients in federated learning. After finetuning, the Attack Success Rate (ASR) decreases to a mere two percent, while accuracy remains consistently high.

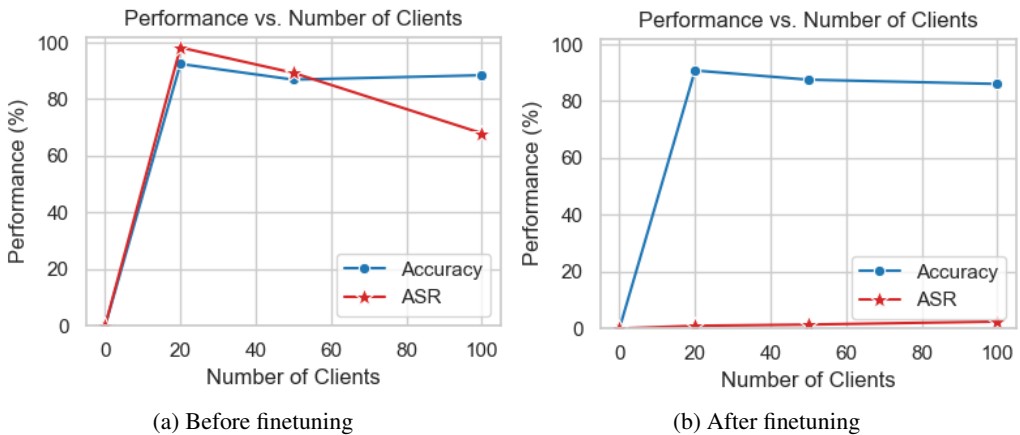

(a) Before finetuning

(b) After finetuning

Figure 7: Impacts (ASR and Accuracy) of the number of clients on the performance of the proposed method.

**Impact of Accuracy of the trustworthy historical base model.** For this examination, we specifically chose clean historical global models from our experiment with accuracies of 63.76%, 71.31%, 74.52%, 80.06%, and 88.62%. We tested these models with our method on the backdoored global model, which had an accuracy of 92.41% and an ASR of 97.58%. Fig. 8 demonstrates that our method continues to be effective even when the model's performance is low, indicating the importance of maintaining a historical clean model.

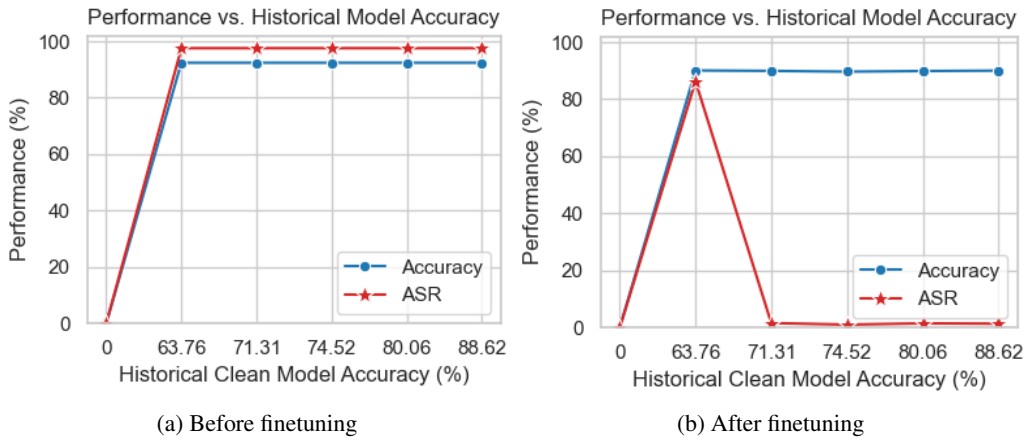

(a) Before finetuning           (b) After finetuning

Figure 8: Impacts (ASR and Accuracy) of accuracies of historical clean models on the performance of the proposed method.

## A.5    RESULTS OF BACKDOOR MODEL DETECTION

Table 4: Results of backdoor model detection on a selection of models with the threshold values set at an MSE of 10 and a REASR of 80. We observe that our method excels in detecting malicious models across most cases, with the exception of the CIFAR-10 dataset under the DBA attack, where Neural Cleanse achieved the best result.

| Dataset | Method | Detection Rate (%) | |
|---|---|---|---|
| | | SBA | DBA |
| MNIST | NC | 87 | 90 |
| | TABOR | 87 | 94 |
| | DeepInspect | **100** | 90 |
| | Ours | **100** | **100** |
| CIFAR10 | NC | 47 | **94** |
| | TABOR | 80 | 80 |
| | DeepInspect | 70 | 77 |
| | Ours | **97** | 87 |
| GTSRB | NC | 60 | 50 |
| | TABOR | 60 | 47 |
| | DeepInspect | 47 | 50 |
| | Ours | **100** | **100** |

