# OpenReview forum: "Efficient Backdoor Mitigation in Federated Learning with Contrastive Loss"
_ICLR.cc/2024/Conference — Submitted to ICLR 2024_

### Official Review · Reviewer_58k3 · 2023-10-26

**Soundness:** 2 fair
**Presentation:** 2 fair
**Contribution:** 2 fair
**Rating:** 3
**Confidence:** 3

**Summary:**

The paper proposes a new method to defend against backdoor attacks and reverse engineer the trigger in federated learning. The proposed method leverages contrastive loss to recover the trigger by comparing the feature outputs of a clean model and the suspicious model. Once the trigger is generated, clean images and images with triggers are fed into the model to check if it has backdoor. If the model has backdoor, techniques similar to Neural Cleanse will be used to remove the backdoor. Experiments show that the proposed method good against two backdoor attacks on three image datasets compared to several centralized backdoor defense method.

**Strengths:**

Strengths:

- The paper proposes a novel method to reverse engineer triggers in federated learning.

- The method outperforms several reverse engineering methods designed for centralized setting in the experiments.

- The idea is clearly described with figures.

**Weaknesses:**

Weaknesses:

- For the goal of defending against backdoor attacks, the assumptions of the method seem too strong: 1) There exists a clean historical model with reasonable performance. 2) There exists a set of clean data to do sanity check. If training from scratch and the attack starts from the first round, where does the clean historical model come from. The goal of federated learning is to avoid data sharing and protect privacy. If there exists a set of clean data for the server, it does not align with the goal of federated learning.

- In the experiments, only centralized backdoor defense methods are compared with the proposed method. There have been so many backdoor defense methods proposed for federated learning the past years. They should be included in the experiments.

Minor:

- Page 2, Paragraph 3: "..., and let the poised image go through...". Maybe poisoned image?

**Questions:**

See the above part.

---

### Official Review · Reviewer_pCAH · 2023-10-31

**Soundness:** 2 fair
**Presentation:** 2 fair
**Contribution:** 2 fair
**Rating:** 3
**Confidence:** 3

**Summary:**

This work introduces a novel approach to detecting and mitigating backdoor attacks in federated learning by utilizing a contrastive learning loss. The contrastive loss is designed to discern differences in responses between a clean model and a backdoored model, ultimately generating the exact trigger. Once the trigger is generated, it is used to detect and cleanse the backdoored model. This technique has been demonstrated to be effective against two attack scenarios with different architectural backbones in a standard federated learning setting. The comparative analysis reveals that it is more effective than existing methods.

**Strengths:**

1. The design of the contrastive objective is logically sound.
2. The consideration of both SBA and DBA attack scenarios enhances the comprehensiveness of the study.
3. The use of a standard federated learning setup and the analysis of the impact of the number of clients add value to the research.
4. The method's non-requirement for iteration across different classes may contribute to faster execution compared to other iterative approaches.

**Weaknesses:**

1. The proposed method has a significant limitation concerning the existence of a clean model. The authors assert that the public model should be from a verified old version of the global model or a renowned pre-trained model that lacks a backdoor. This assumption is practical and bold. In the former case, addressing how to perform the verification when backdoors might occur early in training is not specified. In the latter case, the restriction of needing a publicly available model limits the application. Furthermore, the appendix results support that the defense fails if the clean model's accuracy performance is too poor.
2. Another limitation is the method's capability to defend only small patches, as noted in the appendix.
3. The experimental details regarding the contrastive loss function and the backdoor attack are lacking. Specifics like the target class (which class?) in the backdoor attack and how the Attack Success Rate (ASR) is calculated (e.g., whether it excludes images from the target class) should be provided.
4. While the design itself is logical, there is no guarantee that the final convergence point is exactly (or close to) the real backdoor trigger. The paper should discuss how to guarantee such a strong connection.
5. There isn't strong evidence provided to suggest that this method is only suitable for Federated Learning models. The paper should elaborate on why this approach is particularly suited for Federated Learning models.
6. The paper does not discuss performance against knowledgeable (adaptive) attackers.
7. The method assumes that attackers are effective and stealthy, but it lacks a clear definition of "stealthy." The paper should also consider scenarios where the attacker's objective is not necessarily about being stealthy.
8. Some missing literature should be involved into discussion:

[1] Ozdayi et al. - Defending against Backdoors in Federated Learning with Robust Learning Rate

[2] Zhang et al. - Flip: A provable defense framework for backdoor mitigation in federated learning

[3] Rodríguez-Barroso et al. - Backdoor attacks-resilient aggregation based on Robust Filtering of Outliers in federated learning for image classification

**Questions:**

Can the author elaborate on how to use a sanity-check dataset to do the validation in section 2.1, what’s the criteria?

---

### Official Review · Reviewer_LVhf · 2023-10-31

**Soundness:** 2 fair
**Presentation:** 2 fair
**Contribution:** 2 fair
**Rating:** 3
**Confidence:** 5

**Summary:**

The paper introduces a new method to identify and mitigate backdoor attacks in federated learning setups, utilizing a self-supervised contrastive learning loss. Differing from existing techniques like Neural Cleanse, the proposed method applies contrastive loss broadly to detect triggers in compromised models, comparing the last-layer feature outputs with a pre-saved clean model. The reverse-engineered trigger helps in patching the affected global model. The technique showcased better performance and time efficiency in detecting backdoors in federated learning, compared to Neural Cleanse, TABOR, and DeepInspect, which struggled outside centralized learning settings.

**Strengths:**

Novel loss function to mitigate backdoors.

**Weaknesses:**

The related work section of the paper significantly falls short in addressing the existing literature on backdoor attacks within federated learning environments. Notably absent is any mention or discussion of noteworthy defenses such as KRUM, FoolsGold, FLAME, and FLTrust, which are designed to thwart backdoor injections into the global model. Moreover, the paper overlooks defenses like RLR that employ modified loss functions to avert backdoor injections. There is a significant lack of comparative analysis with these established works, which undermines the paper's contribution to the existing body of research on this subject.

The paper also overlooks evaluating the proposed defense against state-of-the-art attacks such as Neurotoxin, 3DFed, etc.

The datasets employed in the experimental assessment are quite basic. Additionally, the evaluation overlooks the non-IID data distributions from the LEAF benchmark. The functionality of backdoor trigger reverse-engineering and unlearning in real-world non-IID scenarios remains unclear.

**Questions:**

Appropriately address related work in federated learning both for attack and defense.

---

### Official Review · Reviewer_7bgg · 2023-11-01

**Soundness:** 3 good
**Presentation:** 1 poor
**Contribution:** 1 poor
**Rating:** 1
**Confidence:** 4

**Summary:**

While deep learning thrives in various domains, concerns around neural backdoor attacks in federated learning systems persist. Addressing this, the paper presents a novel technique that leverages a contrastive loss within a self-supervised learning framework to efficiently identify and counter backdoors. Unlike traditional methods, this approach employs a clean pretrained model and adapts perturbations to match potential backdoor triggers. The method's efficacy lies in the contrastive loss's ability to distinguish between clean and potentially compromised models. Experiments demonstrate that this technique not only rapidly detects and mitigates backdoor threats but also integrates smoothly into federated learning without needing changes to aggregation methods or accessing local training data.

**Strengths:**

1.  The paper introduces a novel technique using contrastive loss within a self-supervised learning framework, which has proven to efficiently detect and neutralize backdoors. Moreover, this method seamlessly integrates with existing federated learning systems without the need for modifications to aggregation methods or accessing local training data.

2. Through experimental evaluations, the paper's proposed method showcased better performance in mitigating backdoors in federated learning compared to other popular defense strategies. It offers a robust solution that sidesteps challenges like iterating through every class in the dataset or requiring extensive computational resources, setting it apart from conventional defense mechanisms.

**Weaknesses:**

1. Writing needs to be improved. Below examples are merely from the abstract.

a. "Our method compares the last-layer feature outputs of
a potentially affected model with these from a clean one preserved beforehand to
reconstruct the trigger under the guidance of the contrastive loss." This sentence is hard to read and hard to understand.

b. "If the global model is free from backdoors, the Contrastive loss
will lead to either a blank trigger or one with random pattern. " "C" should be in lower case.

c. Since you are claiming a fact (your methods and experimental results), it is recommended to use the simple present tense.

2. Reviewer didn't see any necessary connection between this method and the federated learning, though the experiments are down under the scenario of federated learning. There are no challenges that are brought by the federated learning this paper solves. Neither the method is dedicated to the federated learning such that it considers the heterogeneity problems or the communication costs.

3. The existence of M_c is not well justified. Why does such model exists in the federated scenario. Does the server owns it or the user owns it? If the server, how does the model trained.

4. Why this paper achieves better efficiency is not well justified. Do previous methods have a sequential dependency over the class such that they has to proceed in certain order or it were just the implementation?

5. The contribution of this paper is limited. It simply add a contrastive loss to the reverse engineering problem and perform experiments to show the performance without any deep insights.

**Questions:**

1. "In contrast to existing reverse engineering techniques, such as Neural Cleanse, which involve iterating through each
class in the dataset, we employ the contrastive loss as a whole to identify triggers
in the backdoored model." This sentence is not clear. What does it mean by as a whole? Is it compute the loss as one batch or there is no sequential dependency?

2. "where the trigger image is updated by
comparing the feature outputs of the clean model and the suspicious model through THE contrastive
loss." What is THE contrastive loss?

---

### Meta-Review · Area_Chair_Cddx · 2023-12-09

**Metareview:**

This work studied the backdoor defense in federated learning.

Four detailed reviews proposed lots of important concerns about this work, but no responses. This work is not ready for publication at ICLR.

**Justification For Why Not Higher Score:**

see above

**Justification For Why Not Lower Score:**

n/a

---

### Decision · Program_Chairs · 2024-01-16

Reject